# Application of Antisense Oligonucleotides as an Alternative Approach for Gene Expression Control and Functional Studies

**DOI:** 10.3390/ijms262110524

**Published:** 2025-10-29

**Authors:** Amelia Szukowska, Magdalena Żuk, Julia Sztompke, Bartosz Bednarz, Urszula Kaźmierczak

**Affiliations:** 1Department of Cellular Molecular Biology, Faculty of Biotechnology, University of Wroclaw, F. Joliot-Curie 14A, 50-383 Wroclaw, Poland; 330618@uwr.edu.pl (A.S.); 339011@uwr.edu.pl (J.S.); 2Department of Genetic Biochemistry, Faculty of Biotechnology, University of Wroclaw, Przybyszewskiego 63/77, 51-148 Wroclaw, Poland; magdalena.zuk@uwr.edu.pl; 3Department of Chemical Biology, Faculty of Biotechnology, University of Wroclaw, Joliot-Curie 14A, 50-383 Wroclaw, Poland

**Keywords:** ASO, ODN, OLIGO, gene therapy, epigenetics, plant and animal modification, gene silencing, gene activation

## Abstract

Antisense oligonucleotides (ASOs) are short, synthetic DNA fragments that offer a powerful means of modulating gene expression. By leveraging endogenous regulatory pathways, ASOs enable precise control over gene activity at multiple levels, including genomic DNA, transcription, RNA processing, and translation. Their applications span basic research and translational science, ranging from the generation of epigenetically modified organisms as potential GMO alternatives to the development of therapies for rare or treatment-resistant diseases. This review highlights the molecular mechanisms of ASO action, design and modification strategies, and delivery approaches across diverse cell types. Future directions include elucidating detailed molecular pathways, optimizing experimental conditions, and enhancing the persistence of therapeutic effects. Overall, ASOs represent a versatile and innovative tool in functional genomics, with broad implications for molecular biology, biotechnology, and medicine.

## 1. Introduction

Antisense oligodeoxyribonucleotides (ASOs, ODNs) are short, synthetic, single-stranded DNA molecules designed to specifically bind selected RNA or DNA sequences through Watson–Crick complementary base pairing [1,2]. Their effects on gene expression are multifaceted, affecting various stages of the process, including transcription, RNA maturation, translation, and mRNA degradation [3]. Depending on the binding site, ASOs can modulate gene expression either by upregulating or downregulating activity, without introducing permanent changes to the genome. Thus, they represent a precise and reversible tool for regulating gene activity [4,5].

Antisense oligonucleotide (ASO) technology originated in the late 1970s with the pioneering work of Paul Zamecnik and Mary Stephenson, who first demonstrated that a synthetic oligonucleotide could specifically inhibit viral RNA expression [6]. This discovery established the “antisense” concept—regulating gene expression by hybridizing to target mRNA and modulating its translation or inducing degradation via RNase H [2]. Early research faced major challenges such as oligonucleotide instability and inefficient cellular delivery, which were later overcome through advances in chemical modifications, including phosphorothioate backbones and 2′-O-methyl or phosphorodiamidate morpholino oligomer (PMO) substitutions [7]. These developments significantly improved ASO stability, specificity, and bioavailability, paving the way for therapeutic applications. The first ASO drug, fomivirsen, was approved in 1998, followed by clinically successful agents such as nusinersen and eteplirsen, confirming the potential of this platform in treating genetic disorders [8].

In medicine, ASOs have been applied to correct defective transcripts, enabling the synthesis of functional proteins [9,10,11,12], as well as in therapies targeting diseases caused by pathogens, such as bacteria or fungi in eukaryotic cells [13,14,15]. The success of studies in animal models has inspired experiments with ASOs in plant systems, revealing their potential to induce transgenerational epigenetic modifications. Consequently, ASOs represent a promising tool for both basic research and biotechnological and therapeutic applications [16].

## 2. Gene Expression Control by ASOs Utilizing Naturally Occurring Mechanisms

Antisense oligonucleotides regulate gene expression by engaging endogenous cellular mechanisms, including steric blockade, triplex DNA formation, and RNA interference pathways. As exogenous molecules, ASOs can elicit host defense responses, paralleling natural strategies in plants, where viral infection triggers virus-induced gene silencing (VIGS) or microRNA-mediated VIGS (MIR-VIGS) to limit pathogen replication [17,18]. In animals, small RNA pathways, such as siRNA and miRNA, similarly provide antiviral defense while orchestrating precise gene regulation.

### 2.1. RNA-Level Control—Regulation of Transcription and Translation

The mechanisms employed by antisense oligonucleotide (ASO) technology resemble those triggered by endogenous sequences. They exploit natural pathways involving endogenous non-coding small RNAs, such as microRNAs (miRNAs) and small interfering RNAs (siRNAs) [19,20]. A key biological mechanism utilized in the regulation of gene expression is RNA interference (RNAi) and, in some cases, RNA activation (RNAa) [21].

MicroRNAs are single-stranded RNA molecules that do not require full complementarity to their target sequences, enabling them to regulate the expression of multiple genes. miRNA precursors are transcribed by RNA polymerase II or III and adopt a characteristic hairpin structure consisting of a stem, a loop, and flanking regions [19,22]. During maturation, they are processed by RNase III enzymes—primarily DROSHA and DICER—yielding mature miRNAs of approximately 21–24 nucleotides in length. In animal cells, maturation occurs both in the nucleus and cytoplasm, whereas in plants the entire process takes place in the nucleus [19,23].

Animal miRNAs typically bind to the 3′ untranslated region (3′UTR) of mRNA, thereby repressing translation. In contrast, plant miRNAs exhibit greater complementarity to their target sequences, usually binding within the coding sequence (CDS) or the 5′ untranslated region (5′UTR), which generally results in mRNA degradation [24,25]. Less frequently, repression may occur at the level of transcriptional initiation or elongation [26].

By contrast, siRNAs are generally fully complementary to a single, specific mRNA sequence and possess a double-stranded structure. They most often originate from exogenous double-stranded RNA (dsRNA), such as viral RNA. DICER processes dsRNA into fragments of approximately 19 nucleotides with unpaired ends. Both miRNAs and siRNAs are incorporated into the RNA-induced silencing complex (RISC), which includes proteins such as DICER and ARGONAUTE (AGO). Within this complex, one strand (the so-called passenger strand) is degraded, while the other (the guide strand) remains active and directs recognition of the target mRNA based on sequence complementarity [18,20].

### 2.2. Mechanisms of Action of Synthetic Oligonucleotides

Synthetic oligonucleotides utilize mechanisms analogous to those of endogenous small RNAs, but operate with greater precision and stability [3,22,27]. Depending on their design and target sequence, antisense oligonucleotides (ASOs) can act through several pathways (Figure 1).

Steric blocking—ASOs physically obstruct mRNA or hnRNA molecules, preventing further maturation (e.g., capping, poly(A) tail synthesis) or transcript function [28].Splicing modulation—ASOs can bind to splice sites within pre-mRNA, influencing exon inclusion and alternative splicing, ultimately altering the repertoire of protein isoforms produced [29]. For instance, ASO binding to splicing enhancers may block the recruitment of splice-promoting factors, resulting in exon skipping, whereas binding to splicing silencers can prevent repressor binding, thereby relieving inhibition and promoting exon inclusion [30].Poly(A) tail degradation—ASOs may shorten the poly(A) tail, thereby reducing mRNA stability and accelerating transcript degradation [31].Nonsense-mediated mRNA decay (NMD) pathway—ASO binding to pre-mRNA can induce the formation of transcripts containing premature termination codons (PTCs), resulting in aberrant mRNAs that are subsequently degraded via the NMD pathway [32].RNA interference (RNAi) pathway—by mimicking siRNAs or miRNAs, ASOs can be incorporated into the RNA-induced silencing complex (RISC), leading to selective mRNA degradation or translational repression [2].RNase H pathway—specific to DNA-based oligonucleotides, in which RNase H recognizes DNA-RNA hybrids and degrades the RNA strand. It occurs primarily in the nucleus and constitutes one of the principal applications of antisense oligonucleotides [33].RNA activation (RNAa) pathway—some ASOs can activate gene expression through interactions with promoters, enhancers, translation start sites, or transcription factors. RNAa typically involves ~21-nt double-stranded RNAs resembling siRNAs, which, instead of repressing expression, enhance transcriptional activity of the target gene [34].Anti-miRNA activity (antagomirs)—ASOs can bind endogenous miRNAs, preventing their interaction with target mRNAs. This results in derepression of genes previously inhibited by the miRNA [35].

### 2.3. Regulation of Translation by Antisense Oligonucleotides

As previously noted, ASOs can also regulate gene expression by either repressing or enhancing translation (Figure 2). Mechanisms leading to translational inhibition include ribosome stalling through steric blocking, interference with the 5′ cap, or disruption of the poly(A) tail, thereby preventing the assembly of structures required for translation initiation [36].

Conversely, ASOs may bind to upstream open reading frame (uORF) sequences, blocking their translation and thereby promoting translation of the main open reading frame (mORF). Similarly, ASO interaction with other translation-suppressive elements, such as translation inhibitory elements (TIEs), can alleviate their repressive effects, allowing more efficient translation of the target mRNA [37].

### 2.4. DNA-Level Control—Epigenetic Imprinting

The application of synthetic antisense oligonucleotides (ASOs) enables the induction of epigenetic modifications in a sequence-specific manner without altering the primary genomic sequence [38]. Epigenetic regulation of gene expression plays a pivotal role in genome function, allowing precise adjustment of gene activity to environmental conditions, developmental stages, and tissue types [39]. An important advantage of induced epigenetic changes is their heritability across generations, making ASO-based technologies an attractive alternative to classical genetic modification methods [40].

One of the core mechanisms of this regulation involves chemical modification of DNA and chromatin structure, including cytosine methylation, histone modifications, and chromatin domain remodeling [41] (Figure 3).

Chemical modifications such as DNA methylation (predominantly 5′ cytosine methylation–5-mC and N6-methyladenine–6mA), as well as histone acetylation, methylation, or ubiquitination, influence chromatin condensation and the accessibility of promoter sequences to the transcriptional machinery [42]. In plants, DNA methylation constitutes the dominant form of epigenetic regulation, whereas in mammals histone modifications play a more prominent role [43].

These processes can be further supported by the RNA-induced transcriptional silencing (RITS) complex, analogous to the well-characterized RNA-induced silencing complex (RISC) (Figure 3). RITS contributes to transcriptional repression by promoting DNA methylation and histone modifications, ultimately leading to heterochromatin condensation [44].

### 2.5. Epigenetic Modulation by Antisense Oligonucleotides

Oligonucleotides can also induce epigenetic modifications by binding to specific genomic regions and forming triplex DNA structures, which serve as recruitment signals for methyltransferase enzymes [45]. This results in localized methylation of selected sequences, potentially leading to stable repression or activation of the target gene. For example, studies in flax (*Linum usitatissimum*) have demonstrated that methylation changes at CG, CHG, and CHH sites can result in both repression and enhancement of target gene transcription [46]. In all these methylation motifs, cytosine is methylated as follows: in CG, cytosine is methylated before guanine; in CHG, before any nucleotide followed by guanine; and in CHH, before two nucleotides other than guanine.

Such modifications lead to reorganization of chromatin structure, affecting the accessibility of promoter and enhancer regions to transcription factors. Importantly, even single methylation changes can trigger nucleosome repositioning, altering local chromatin architecture and modifying interactions between distal genomic regions, including within topologically associating domains (TADs) [47]. These domains play a crucial role in the spatial organization of chromatin and the coordination of gene expression. Consequently, ASO-induced epigenetic changes may affect the activity of entire gene clusters rather than individual loci [48].

Furthermore, ASOs can be designed to target transcripts encoding DNA methyltransferases, such as DNMT1 or DNMT3, resulting in their silencing. This may lead to a global reduction in genomic methylation levels, which has potential therapeutic applications, particularly in conditions characterized by aberrant hypermethylation (e.g., certain cancers, neurodegenerative diseases, and even autism spectrum disorders) [49,50,51].

In the context of plant biotechnology, epigenetic regulation using ASOs shows considerable potential for improving agronomic traits. Studies in transgenic rice (*Oryza sativa*) and potato (*Solanum tuberosum*) have demonstrated that ASO-induced methylation modifications can significantly increase biomass, enhance drought resistance, and boost crop yields by up to 50% [52].

## 3. Design and Chemical Modifications of ASOs for Optimized Biological Effect

The design and chemical modification of ASOs are critical for achieving optimal biological outcomes. Selecting the appropriate target region, sequence, and chemical modifications enhances the efficiency of gene expression modulation, minimizes off-target effects, and improves ASO stability and bioavailability, which is essential for both basic research and therapeutic applications.

### 3.1. ASO Design

The length of an antisense oligonucleotide (ASO) sequence is a critical factor determining the stability of ASO–RNA or ASO–DNA duplexes. Typically, ASO sequences do not exceed 25 nucleotides, allowing high complementarity while minimizing nonspecific interactions. ASOs may exhibit off-target effects, meaning unintended binding to RNA sequences other than their primary target [53]. Such interactions can result in undesired mRNA degradation or modulation of gene expression, potentially causing cellular toxicity or functional disturbances [54]. Off-target effects are particularly relevant at high ASO concentrations or when using less selective chemical modifications. Current strategies to minimize these effects include optimizing ASO sequences, employing chemical modifications that enhance binding specificity, and conducting thorough bioinformatic and experimental evaluations in both cell-based and animal models.

Nucleotide composition plays a crucial role in determining ASO hybridization efficiency. Sequences with a GC content above 55% generally demonstrate stronger binding, while the presence of specific motifs—such as CTCT, GCCA, CCAC, TCCC, or ACTC—can further enhance duplex stability [55]. Longer ASOs, however, are more prone to forming secondary structures, such as hairpins, which can additionally stabilize the oligomer [56,57]. Consequently, predicting the secondary structure of the target mRNA is essential for optimal ASO design. Bioinformatic tools, such as mfold, enable the assessment of minimum free energy (ΔG) and potential RNA conformations, whereas sfold can identify transcript regions most favorable for hybridization. More detailed information on this topic can be found in our previous review [58].

The direction, mechanism, and overall efficacy of ASO-mediated modulation critically depend on the precise selection of the target region within the DNA sequence or transcript [55,56]. Target accessibility largely determines hybridization efficiency, with spatially exposed mRNA regions being more susceptible to ASO binding [3]. In practice, coding sequences are frequently chosen, particularly the translation initiation site, which exhibits high bioavailability due to minimal secondary structure. Hybridization at this site disrupts the recruitment of initiation factors or the scanning of the mRNA by the ribosomal small subunit, effectively inhibiting translation [59,60].

Alternative ASO targets include the 5′ and 3′ untranslated regions (UTRs), which harbor critical regulatory elements. Hybridization to these regions can either downregulate gene expression or enhance transcript stability, thereby increasing mRNA levels [57,61]. Additionally, targeting splice sites within pre-mRNA can interfere with proper transcript processing, leading to the formation of aberrant mRNA isoforms that are subsequently degraded.

ASOs can also be directed to DNA sequences to induce epigenetic modifications, such as DNA methylation or chromatin remodeling [61,62]. Promoter regions, both distal and proximal, are common targets as they contain diverse regulatory motifs, including silencers, enhancers, *cis*-regulatory elements, and transcription factor binding sites, representing potential hybridization sites for ASOs [57,61].

### 3.2. Chemical Modifications

Due to their negative charge, antisense oligonucleotides (ASOs) exhibit limited cellular membrane permeability. Additionally, they are susceptible to degradation by exo- and endonucleases and may interact with intracellular proteins, which can reduce their activity and potentially induce cytotoxicity [63].

To increase stability, bioavailability, and efficacy, ASOs are chemically modified. These modifications increase resistance to nucleases, prolong half-life, strengthen binding to the target nucleic acid, and reduce nonspecific toxicity [56]. Depending on their location, chemical modifications are classified into three generations (Table 1).

#### 3.2.1. First Generation—Phosphorothioate Backbone Modifications

Phosphorothioates (PS)—this modification involves replacing an oxygen atom in the phosphate backbone with a sulfur atom [64]. PS modifications are most commonly applied at the ends of ASOs in both in vitro and in vivo studies [57]. They increase stability but reduce the melting temperature of the ASO–mRNA heteroduplex by approximately 0.5 °C per modified nucleotide.Methylphosphonates—these modifications replace one of the oxygen atoms in the phosphate group with a methyl group. This results in the ASO molecule losing its negative charge, which enhances stability in biological environments but simultaneously decreases solubility and membrane permeability. Nevertheless, methylphosphonates can be internalized via endocytosis. A significant limitation of their use is the inability to activate RNase H, which precludes their application in strategies requiring target RNA degradation [65].

#### 3.2.2. Second Generation—Sugar (Ribose) Modifications

2′-O-methyl (2′-OMe) and 2′-O-methoxyethyl (2′-MOE)—these are the most commonly used sugar modifications, often combined with phosphorothioate (PS) modifications in gapmer designs [56]. They enhance ASO stability; however, due to the absence of the 2′-OH group, they do not activate the RNase H pathway [66].Hexitol nucleic acids (HNA)—these are synthetic nucleic acid analogs in which the conventional ribose sugar is replaced by a six-membered hexitol ring. This structural alteration of the sugar backbone increases resistance to enzymatic degradation by nucleases and improves the thermal stability of HNA–DNA and HNA–RNA hybrids. HNAs do not induce RNase H cleavage but can effectively block splicing and translation processes [67,68].

#### 3.2.3. Third Generation—Base Analogs and Modified Furanose Rings

Peptide Nucleic Acids (PNA)—synthetic DNA analogs in which the sugar-phosphate backbone is replaced by a pseudopeptide chain composed of repeating N-(2-aminoethyl)-glycine units. Nitrogenous bases are attached to this backbone in a manner analogous to natural nucleic acids, enabling specific hybridization with complementary DNA or RNA. PNAs are electrically neutral and exhibit enhanced resistance to nucleases and proteases. Their mode of action involves inhibition of translation or splicing, as well as modulation of transcription [67,69].Locked Nucleic Acids (LNA)—synthetic nucleotide analogs in which the ribose ring is chemically “locked” through the introduction of an additional methylene bridge (-CH_2_-) connecting the 2′-O and 4′-C atoms of the furanose ring. This modification rigidifies the ribose conformation, increasing duplex stability and hybridization affinity. LNAs markedly enhance oligonucleotide resistance to enzymatic degradation both in vitro and in vivo. They are used to suppress mRNA expression by interfering with splicing or obstructing ribosomal translation. When incorporated into gapmer constructs, LNAs can indirectly activate target mRNA degradation through RNase H [67,69].Phosphorodiamidate Morpholino Oligomers (PMO)—synthetic oligonucleotide analogs in which the conventional phosphodiester backbone is replaced by an uncharged phosphorodiamidate backbone and the ribose is substituted with a morpholine ring [70]. PMOs exhibit high stability in vitro and in vivo due to strong resistance to nucleases and proteases. They do not activate RNase H but function by blocking translation initiation and disrupting pre-mRNA splicing [67]. Being electrically neutral, their cellular uptake is highly limited; thus, delivery is enhanced by conjugation with cell-penetrating peptides (CPPs), such as arginine-rich peptides (ARPs), which significantly improve cellular internalization and functional efficacy [71].

### 3.3. Gapmers and Mixmers—Chimeric Oligonucleotides

Gapmers consist of a central “gap” of approximately 10 phosphorothioate (PS) nucleotides, flanked at the 5′ and 3′ ends by roughly five modified nucleotides (second- or third-generation modifications) [72,73]. This configuration allows RNase H to access the cleavage site, recognize the target and degrade the mRNA, while simultaneously providing protection against exonucleases [56]. Gapmers enable the degradation of target nucleic acids using modifications that would otherwise not activate RNase H due to the absence of the 2′-OH group in the ribose moiety.Mixmers contain alternating modified and natural nucleotides (e.g., LNA/DNA). They are designed primarily to block translation, typically targeting the 5′ untranslated region (5′ UTR) of mRNAs [74].

## 4. Methods of ASO Delivery

Effective delivery of antisense oligonucleotides (ASOs) into cells is crucial for their applications in both basic research and therapeutic contexts [16,75]. However, ASO delivery faces numerous challenges related to cellular barriers as well as the physicochemical properties of the molecules themselves. In animal cells, the negatively charged ASOs are repelled by the cell membrane, and a portion of the oligonucleotides is degraded within endosomes or digested by nucleases, limiting their efficacy. Also plant cell walls possess a net negative surface charge; however, efficiency depends on the species, tissue type, and environmental conditions [76,77].

### 4.1. Delivery of ASOs into Animal Cells

As with most intracellular therapeutics, antisense oligonucleotides (ASOs) are predominantly delivered encapsulated within cationic liposomes or lipid nanoparticles, enabling efficient fusion with the negatively charged plasma membrane via endocytosis [78,79]. Widely employed carriers include N-[1-(2,3-dioleoyloxy)propyl] N,N,N-trimethylammonium chloride (DOTMA) and N-[1-(2,3-dioleoyloxy)propyl]-N,N,N-trimethylammonium methyl sulfate (DOTAP) [56]. Naked oligonucleotides are less commonly used due to charge repulsion, reduced endocytosis efficiency, and heightened vulnerability to nuclease degradation. Mechanical approaches such as electroporation or microinjection are effective in vitro but are largely impractical for in vivo applications [56].

ASO uptake is highly tissue-dependent. The liver and kidneys exhibit the highest accumulation and pharmacodynamic response, whereas the brain, heart, and skeletal muscle display comparatively lower uptake [80]. Uptake efficiency is influenced not only by tissue-specific metabolic activity but also by ASO structural properties, including strand type, charge, encapsulation, chemical modifications, and the number and position of such modifications, all of which impact pharmacokinetics and biodistribution [3]. To further enhance delivery precision, ASOs have been conjugated to molecules such as thyroid hormone T3, N-acetylgalactosamine (GalNAc), dendrimers, and various cell-penetrating peptides (CPPs), achieving variable efficacy [80,81,82,83]. Notably, cellular uptake does not always directly predict molecular activity, underscoring the complexity of ASO pharmacodynamics [79].

Efficient delivery to target cells and engagement with intended molecular pathways remain major challenges in oligonucleotide drug development. Consequently, only a limited number of ASO therapeutics have received FDA approval. Nevertheless, ASOs continue to demonstrate significant therapeutic potential, exemplified by their rapid development as antiviral agents against SARS-CoV-2 [84,85].

### 4.2. Delivery of ASOs into Plants

The efficiency of oligonucleotide uptake varies depending on the plant species, developmental stage, tissue type, and the intended biological effect. Different species and tissues differ in cell wall composition and permeability, which affects how well oligonucleotides are absorbed. Younger or actively dividing tissues often take up oligonucleotides more efficiently than mature ones [86].

One of the most commonly used methods for delivering ASOs into whole plants involves their application in sugar-containing solutions. Membrane transporters specific for monosaccharides (e.g., glucose, fructose) and disaccharides (e.g., sucrose, maltose) enable active uptake of ASOs by plant cells, whereas sugar alcohols such as sorbitol or mannitol exhibit limited efficacy [78,87]. Maximum efficiency is observed at concentrations of 200 mM sucrose or glucose. However, for plants sensitive to osmotic stress, such as wheat, cucumber, or *Arabidopsis*, it is recommended to reduce the concentration to 100 mM to avoid detrimental effects on plant health [57].

An alternative approach is foliar spraying, which involves applying an ASO-containing solution directly to the leaf surface. In one study, four-week-old *Arabidopsis thaliana* plants were sprayed twice daily with 700 μM ASOs dissolved in 80 mM sucrose [88]. This technique is simple and non-invasive, with its effectiveness influenced by factors such as ASO concentration, presence of sugars, application timing, and the anatomical and physiological characteristics of the plant. The method demonstrates potential for both greenhouse and field conditions, although environmental factors and leaf structure may limit its efficacy.

Another widely used technique is infiltration, which exploits the presence of stomata on the abaxial (underside) surface of leaves. For plants with large leaves, such as *Nicotiana tabacum*, syringe infiltration is employed [89], whereas for species with small leaves, such as *Arabidopsis thaliana* or flax (*Linum usitatissimum*), vacuum infiltration is utilized. In this method, leaves are submerged in an ASO-containing solution and placed in a vacuum chamber to facilitate molecular entry into the tissues. Following the procedure, leaves are transferred to growth medium for a defined incubation period [61].

Less commonly employed are physical methods, such as microinjection, biolistic delivery, electroporation, or chemical enhancement of cell membrane permeability. Although these approaches can be effective, their application is limited by high invasiveness, the risk of tissue damage, and labor-intensive procedures [78,90,91].

Liposomal technology is utilized for the delivery of ASOs into tissues lacking a cell wall, such as protoplasts or cell suspensions, as well as into tissues with highly permeable cell walls, for example, anthers [92]. In the case of anthers, effective delivery may also be achieved by adding free, unbound ASOs directly to the growth medium, since the presence of phosphates negatively affects their development.

## 5. Applications of ASO Technology

Antisense oligodeoxynucleotides represent a modern tool in molecular biology and medicine, enabling precise regulation of gene activity. While this technology has been extensively developed in animal studies and clinical research, it is increasingly applied in plant research as well.

The first report demonstrating the potential utility of this method in molecular biology and medical therapy dates back to 1978, when it was shown that synthetic ASOs complementary to the Rous sarcoma virus (RSV) could effectively inhibit viral replication in cultures of chick embryo fibroblasts [93]. Since then, ASO-based technology has found broad applications in both animal studies and clinical settings, serving as a valuable complement to conventional therapeutic approaches. Oligonucleotide therapies are now considered a promising alternative for treating diseases that are difficult to manage using traditional therapeutic strategies.

A major breakthrough in the development of oligonucleotide therapies occurred in 2013 with the approval of Mipomersen—the first systemically administered oligonucleotide drug—used for the treatment of familial hypercholesterolemia. By August 2022, sixteen therapies based on short oligonucleotide sequences had been authorized for clinical use [12,14]. Currently approved antisense therapies primarily target rare diseases, such as Duchenne muscular dystrophy (Eteplirsen) and hereditary amyloidosis (Inotersen, Patisiran). Despite high costs—mainly due to the specialized development of drugs for rare conditions—a key advantage of these therapies remains their capacity for precise tailoring to a specific disease type, and even to the individual patient’s needs. Furthermore, ASOs exhibit therapeutic potential in the treatment of diseases caused by pathogens in eukaryotic cells. By selectively binding to pathogen RNA, ASOs can inhibit their replication as well as the expression of genes responsible for antibiotic resistance, representing a promising alternative to conventional antibiotic therapies [13].

The use of ASOs as a tool for specific silencing of selected genes—whether at the DNA level or at the mRNA transcript level—has opened new avenues in plant research. In contrast to time-consuming transgenic techniques, this method allows for rapid and reversible modulation of gene expression, making it particularly valuable for pilot experiments.

Previous studies have demonstrated that ASOs can be employed both to downregulate and upregulate the expression of target genes [61]. For example, oligonucleotides targeting the sugar signaling in barley transcription factor (SUSIBA2) enabled the confirmation of its role in starch biosynthesis [94]. Additionally, the regulation of genes related to plant defense, including modulation of β-glucanase activity, has been reported [62]. Furthermore, the application of chemically modified 2′-deoxy-2′-fluoro-d-arabinonucleic acid antisense oligonucleotides (FANA ASO) targeting *Candidatus Liberibacter asiaticus* genes [95] and antisense oligonucleotides directed against *CsLOB1* in *Xanthomonas citri* subsp. *citri* [96] highlights the potential of this technology in controlling citrus diseases.

Antisense oligonucleotides have also been utilized in studies of secondary metabolism. For instance, silencing of chloroplast protein expression led to a reduction in carotenoid content [89,97], while modification of starch branching enzymes *SBEI* and *SBEIIA* genes in barley resulted in significant alterations of starch structure [87]. Moreover, ASO technology enables the functional analysis of genes regulating developmental processes, such as growth and organogenesis [98].

In most cases, the effects of oligonucleotides in plant cells are transient, persisting for several days. A promising research direction is the use of chemically modified ASOs, which can enhance the stability and durability of gene expression changes. Initial experiments confirm that this approach allows for stabilization of gene expression modulation [16]. Another particularly interesting prospect is the application of ASOs for the selection of transformed plant cells through inhibition of specific sequences, enabling high-throughput screening without the introduction of foreign genes [88].

## 6. Summary and Future Perspectives

Antisense oligonucleotides (ASOs), originally developed as biomedical tools, have evolved into versatile agents with significant potential in both plant research and therapeutic applications [14,16]. Their ability to rapidly, precisely, and reversibly modulate gene expression, combined with high chemical stability, makes them valuable instruments in basic research, biotechnology, and medicine.

ASOs offer several advantages over genome editing technologies, including lower cost, greater flexibility, and faster implementation [58,63]. Their use enables precise functional analysis of genes and transcript-specific regulation, supporting detailed exploration of cellular mechanisms [59,60]. Furthermore, the development of epigenetically modified organisms (EMOs) represents a promising alternative to genetically modified organisms (GMOs), particularly under the stringent regulatory framework of the European Union [99]. However, as epigenetic effects tend to be transient across generations, further research is required to improve their stability and heritability.

For applications in animal systems, specific chemical modifications are often necessary to enhance ASO stability and functionality, enabling correction of defective transcripts and restoration of proper protein synthesis [100]. In addition, silencing efficiency varies depending on the target gene, with those involved in basic metabolism generally being less responsive than genes controlling phenotypic traits [58,63].

## Figures and Tables

**Figure 1 ijms-26-10524-f001:**
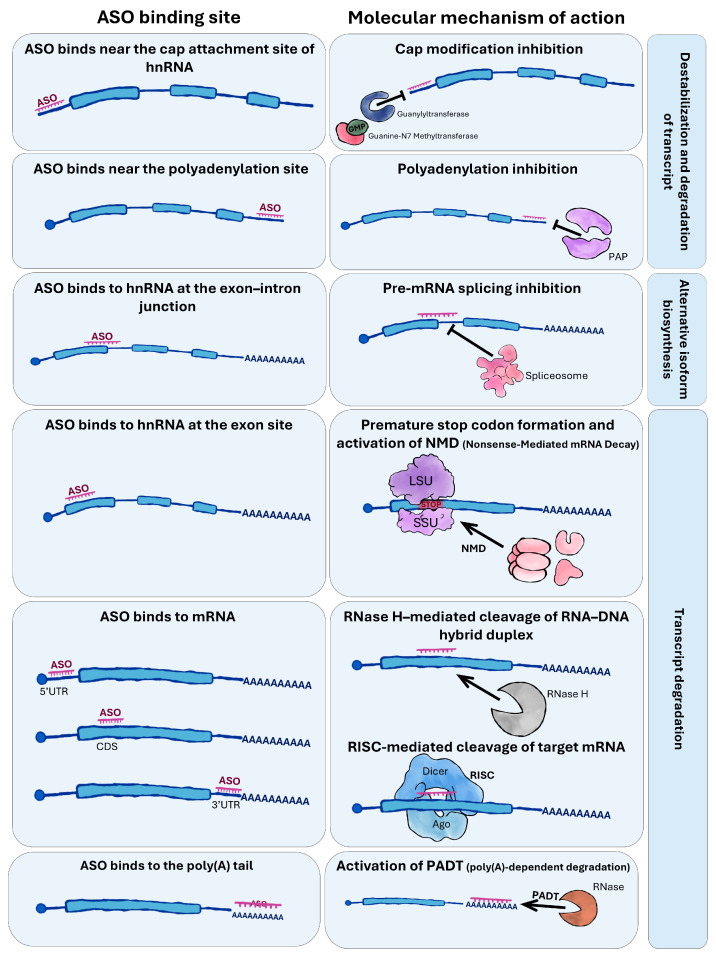
**Mechanisms of action of antisense oligonucleotides (ASO) depending on binding site in mRNA or hnRNA.** ASOs can bind to different regions of RNA, including hnRNA, pre-mRNA, mature mRNA, and poly(A) tails, leading to diverse molecular outcomes. Binding near the cap or polyadenylation (poly(A)) site can inhibit cap formation or polyadenylation, respectively, affecting transcript stability and translation. Binding at exon–intron junctions or within exons can modulate pre-mRNA splicing or induce premature stop codon formation, triggering nonsense-mediated mRNA decay (NMD). ASO binding to mRNA can recruit RNase H to cleave RNA-DNA hybrids or activate poly(A)-dependent degradation (PADT), resulting in transcript degradation. These mechanisms demonstrate the versatility of ASOs in regulating gene expression at multiple post-transcriptional levels.

**Figure 2 ijms-26-10524-f002:**
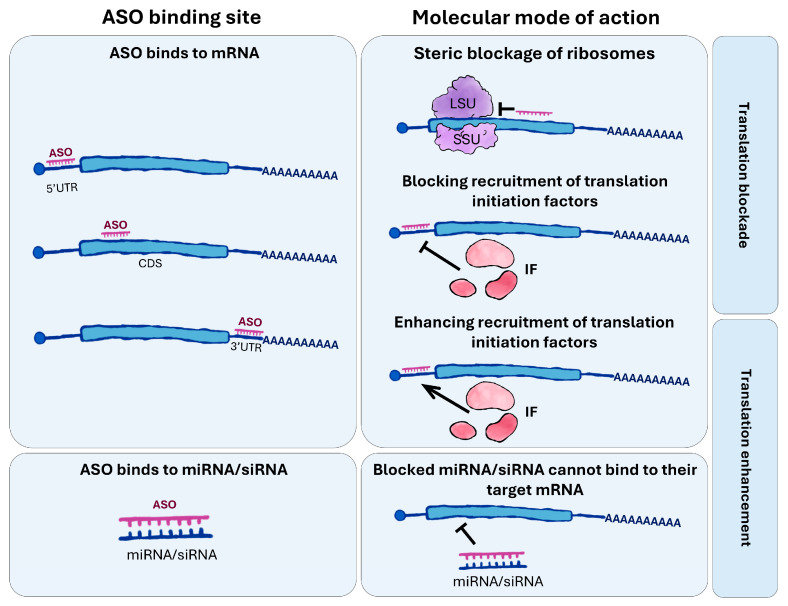
**Mechanisms of action of antisense oligonucleotides (ASOs) depending on binding site.** ASOs can bind to different regions of mRNA. ASOs can target the 5′UTR, coding sequence (CDS), or 3′UTR of mRNAs, modulating translation by impeding ribosome assembly, blocking elongation, or altering the recruitment of translation initiation factors (IFs). By binding to specific regions of the mRNA, ASOs can sterically hinder ribosomal scanning or disrupt secondary structures that are important for efficient translation. In addition to directly interacting with mRNAs, ASOs can sequester regulatory small RNAs, such as miRNAs or siRNAs, preventing their interaction with target transcripts. This inhibition of small RNA-mediated repression can lead to enhanced translation of the corresponding mRNAs. Collectively, these mechanisms allow ASOs to exert precise control over gene expression at the post-transcriptional level.

**Figure 3 ijms-26-10524-f003:**
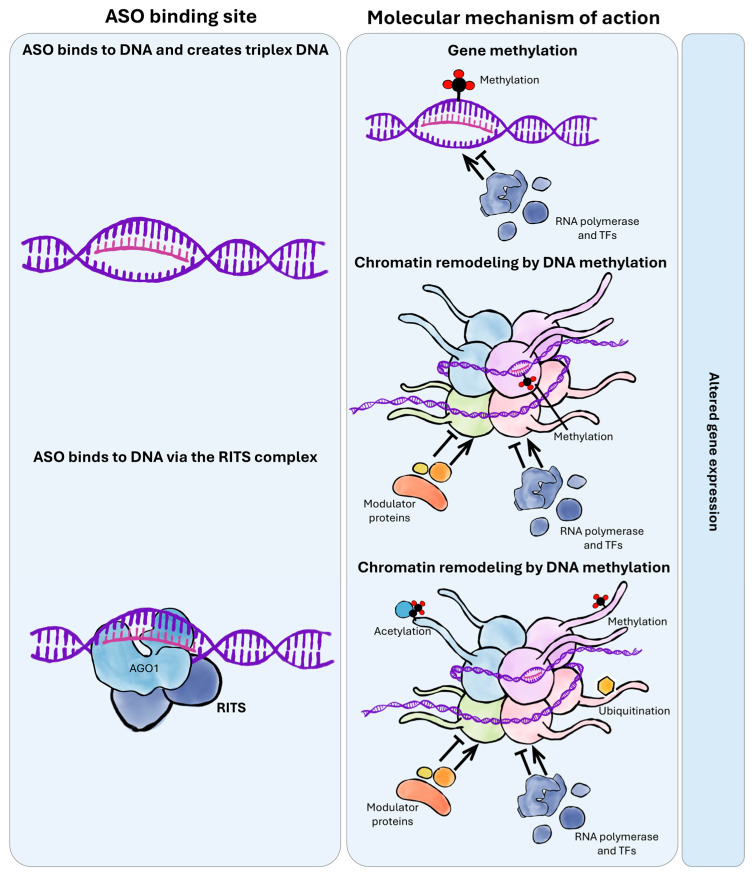
**Mechanisms of action of antisense oligonucleotides (ASOs) binding to DNA.** ASOs can regulate gene expression by interacting directly with DNA. They may form DNA triplex structures, which can interfere with transcriptional machinery or recruit epigenetic modifiers. Alternatively, ASOs can act via RITS (RNA-induced transcriptional silencing) complexes, guiding chromatin-modifying enzymes to specific genomic *loci*. These interactions can result in DNA methylation or chromatin remodeling, ultimately modulating gene expression.

**Table 1 ijms-26-10524-t001:** **ASO chemical modification.** Impact on target sequence binding or ASO molecule stability determined as: – weak effect; + slight; ++ moderate; +++ strong.

Generation	Type of Modification	Chemically Modified Component	Silencing Mechanism	Target Binding	Stability	Adverse Effects
Reference	DNA	No modification	RNase H activity	+	–	Low resistance to nuclease activity: t_1_/_2_ ≈ 20 min; possible nonspecific interactions; toxic degradation products (dNMP)
First	Phosphorothioate (PS)	Replacement of a non-bridging oxygen in the phosphate backbone with sulfur	RNase H activity	–	++ Increased resistance to nuclease degradation; t_1_/_2_ up to 35 h	Interactions with cell surface and intracellular proteins, potentially affecting cell physiology; possible toxicity
Second	2′-O-Methyl (2′-OMe)	Methyl substitution at the 2′ position of ribose	Steric blockade of translational machinery	++	+	Reduced solubility and cellular delivery (restricted to endocytosis)
	2′-O-Methoxyethyl (2′-MOE)	Methoxyethyl substitution at the 2′ position of ribose	Steric blockade of translational machinery	++	+	Reduced solubility and cellular delivery (restricted to endocytosis)
	Hexitol Nucleic Acids (HNA)	Insertion of a methylene group between O4′ and C1′ of the sugar to form a hexose, shifting the nucleobase to the 2′ position	Steric blockade of translational and splicing machinery	++	++ Increased resistance to nuclease and thermal degradation	
Third	Peptide Nucleic Acids (PNA)	Pseudopeptidic backbone (N-(2-aminoethyl)glycine) instead of phosphate; nucleobases attached via methylene-carbonyl linkage	Steric blockade of translational machinery; inhibition of transcriptional elongation, transcription factor binding, and splicing	+++	+++ Increased resistance to nuclease and peptidase degradation; higher stability due to absence of a negatively charged phosphate backbone	Reduced solubility and cellular delivery
	Locked Nucleic Acids (LNA)	2′-O,4′-C-methylene bridge in β-D-furanosyl ribose	Steric blockade of translational and splicing machinery	+++	+++ Significantly enhanced hybridization affinity and thermodynamic stability of duplexes; resistance to nuclease degradation	Potential toxic effects
	Phosphorodiamidate Morpholino Oligomers (PMO)	Six-membered morpholine ring instead of ribose and phosphorodiamidate linkage instead of phosphate backbone	Steric blockade of translational and splicing machinery	+++	+++ Resistance to nuclease and protease degradation	Possible interactions with nucleic acid–binding proteins due to lack of charge

## Data Availability

No new data were created or analyzed in this study. Data sharing is not applicable to this article.

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
