# Peer review of "Application of Antisense Oligonucleotides as an Alternative Approach for Gene Expression Control and Functional Studies"

_ijms, 2025, doi:10.3390/ijms262110524_

Round 1
Reviewer 1 Report
Comments and Suggestions for Authors
1. In the introduction section, provide a more detailed exposition of the developmental backdrop of ASOs technology, encompassing the challenges and constraints encountered during its nascent stages of researchï¼›
2. "Antisense oligonucleotides (ASOs, previously also known as ODNs, hereafter uniformly referred to as ASOs)", and ASOs will be used consistently thereafterï¼›
3. The descriptions in the figure captions are relatively brief. It is recommended to provide more detailed annotations of the key steps for each mechanismï¼›
4. Check and revise the references in the manuscriptï¼›
5. The manuscript should include descriptions of ASOs technology that address issues such as off-target effects and long-term safety.
6. Some paragraphs in the manuscript require refinement. For example, the content in the conclusion section is somewhat repetitive and could be merged and streamlined.
Author Response
Comments1:In the introduction section, provide a more detailed exposition of the developmental backdrop of ASOs technology, encompassing the challenges and constraints encountered during its nascent stages of researchï¼›
Response 1:In accordance with the suggestion, we have enriched the introduction with this information.
Comments 2:"Antisense oligonucleotides (ASOs, previously also known as ODNs, hereafter uniformly referred to as ASOs)", and ASOs will be used consistently thereafterï¼›
The term ODNs, although not used further in the manuscript, is a vital and valuable factor that may assist in searching for methodological information and broadens the search range. For this reason, we believe that this passage should remain unchanged.
Comments 3: The descriptions in the figure captions are relatively brief. It is recommended to provide more detailed annotations of the key steps for each mechanism
Response 3:We note that an editorial oversight occurred in this section. The text intended as an elaboration of the main title under the figures was inadvertently included in the main manuscript text. This has now been corrected to ensure proper placement and clarity of the figure descriptions.
Comments 4: Check and revise the references in the manuscriptï¼›
Response 4: We have updated and, where appropriate, added citations, particularly those referring to our previous work on ASOs. These changes ensure that relevant background and prior findings are properly acknowledged and provide a more comprehensive context for the current study.
Comments 5: The manuscript should include descriptions of ASOs technology that address issues such as off-target effects and long-term safety.
Response 5:Changes, in accordance with the suggestion, have been made in the “ASO Design” chapter.
Comments 6: Some paragraphs in the manuscript require refinement. For example, the content in the conclusion section is somewhat repetitive and could be merged and streamlined.
Response 6:We have shortened the conclusion section. While some information partially repeats content presented earlier in the manuscript, this was done intentionally to emphasise the key aspects and significance of ASO technology. This approach ensures that the main take-home messages are clearly highlighted for the reader
Reviewer 2 Report
Comments and Suggestions for Authors
The article is correct, informative and write very well.
Only the phrase in the lines 483-485 could be rewrite because:
In the page 8, lines 257-260 is write that:
“Due to their negative charge, antisense oligonucleotides (ASOs) exhibit limited cellular membrane permeability. Additionally, they are susceptible to degradation by exo and endonucleases and may interact with intracellular proteins, which can reduce their activity and potentially induce cytotoxicity [50]”
And in the page 8, lines, 261-264 is write that is necessary to “ASOs are chemically modified” to “reduce nospecific toxicity…”
“To increase stability, bioavailability, and efficacy, ASOs are chemically modified. These modifications increase resistance to nucleases, prolong half-life, strengthen binding to the target nucleic acid, and reduce nonspecific toxicity [45] Depending on their location, chemical modifications are classified into three generations (Table 1).”
So, in the page 14, lines 483-485 the phrase:
“Advantages include low synthesis cost, reduced cytotoxicity, and correction of defective transcripts, enabling the production of functional proteins.”
Could be rewrite as:
“Advantages include low synthesis cost and reduced cytotoxicity. However, a special chemical modification on ASOS may be necessary for use in animal cells. This enables the correction of defective transcripts and the production of functional proteins.”
Author Response
Comments: The article is correct, informative and write very well. Only the phrase in the lines 483-485 could be rewrite because: In the page 8, lines 257-260 is write that: “Due to their negative charge, antisense oligonucleotides (ASOs) exhibit limited cellular membrane permeability. Additionally, they are susceptible to degradation by exo and endonucleases and may interact with intracellular proteins, which can reduce their activity and potentially induce cytotoxicity [50]”And in the page 8, lines, 261-264 is write that is necessary to “ASOs are chemically modified” to “reduce nospecific toxicity…”“To increase stability, bioavailability, and efficacy, ASOs are chemically modified. These modifications increase resistance to nucleases, prolong half-life, strengthen binding to the target nucleic acid, and reduce nonspecific toxicity [45] Depending on their location, chemical modifications are classified into three generations (Table 1).” So, in the page 14, lines 483-485 the phrase:“Advantages include low synthesis cost, reduced cytotoxicity, and correction of defective transcripts, enabling the production of functional proteins.”Could be rewrite as:“Advantages include low synthesis cost and reduced cytotoxicity. However, a special chemical modification on ASOS may be necessary for use in animal cells. This enables the correction of defective transcripts and the production of functional proteins.”
Response: The manuscript summary has been updated to reflect the revisions proposed by the reviewer.
Reviewer 3 Report
Comments and Suggestions for Authors
The review is dedicated to the topic of application of antisense oligonucleotides (ASO) in molecular biology, biotechnology, and medicine. The review is written in a very structured and logical manner with informative illustrations. The authors covered all aspects of working with antisense oligonucleotides from the fundamental mechanisms of action to design, delivery, and medical application.
The review is very interesting and useful for the scientific community, but it should be noted that two years ago the authors already published a similar review on the same topic https://doi.org/10.3390/ijms24054466 in the same journal. This situation is not surprising when you are actively working in one direction for which you have funding, but for future publications I recommend authors to pay more attention publishing less similar works. The authors in the current review significantly improved the concept and provided valuable information in the sections devoted to chemical modifications of ASOs, delivery of ASOs into animal cells and applications of ASO technology. At the same time, sections containing information about mechanisms of action, design, and delivery of ASOs into plants share a very high similarity with the previous review.
I think the authors recognize the high degree of similarity between the two reviews, and therefore recommend that the authors rewrite the most similar passages, adding new research and references to the previous review.
Recommendations:
- Line 433: the sentence «Application of Antisense Oligonucleotides as a Tool for Specific Gene Silencing in Plant Research » seems to be a subtitle, but it is not highlighted as a subtitle.
- Lines 228-230: «Sequences with a GC content above 55% generally demonstrate stronger binding, while the presence of specific motifs—such as CTCT, GCCA, CCAC, TCCC, or ACTC—can further enhance duplex stability [4], [44] » authors should remove the reference for their previous review as it does not contain results of original study mentioned here. Alternatively, authors should rewrite and shorten the «Design» section with addition of reference «reviewed in [4]» because all the information was published before.
- Line 251: «ASOs can also be directed to DNA sequences to induce epigenetic modifications, such as DNA methylation or chromatin remodeling [4]» please provide a link to original study but not the review.
- Line 369: «Plant cells exhibit greater susceptibility to ASOs than animal cells, primarily due to their positively charged cell walls, which do not pose a barrier to the negatively charged ASO molecules [4]» please provide a link to original study.
- Line 478: «Promising applications include the generation of epigenetically modified organisms (EMOs) as alternatives to GMOs, which remain tightly regulated in the EU [4].» please provide a link to studies demonstrating generation of epigenetically modified organisms, if possible, and a link to EU regulations.
- Lines 233-236: «Bioinformatic tools, such as mfold, enable the assessment of minimum free energy (ΔG) and potential RNA conformations, whereas sfold can identify transcript regions most favorable for hybridization.» here is a good sentence for reference for the previous review «reviewed in [4]» as it contains more detailed and useful information about the bioinformatic tools.
Author Response
Comments 1: The review is very interesting and useful for the scientific community, but it should be noted that two years ago the authors already published a similar review on the same topic https://doi.org/10.3390/ijms24054466 in the same journal. This situation is not surprising when you are actively working in one direction for which you have funding, but for future publications I recommend authors to pay more attention publishing less similar works. The authors in the current review significantly improved the concept and provided valuable information in the sections devoted to chemical modifications of ASOs, delivery of ASOs into animal cells and applications of ASO technology. At the same time, sections containing information about mechanisms of action, design, and delivery of ASOs into plants share a very high similarity with the previous review. I think the authors recognize the high degree of similarity between the two reviews, and therefore recommend that the authors rewrite the most similar passages, adding new research and references to the previous review.
Response1: The aim of this review was to expand upon our previous publication by including the application of ASOs in animals and by presenting the information from the earlier article from a different perspective and in a more organised manner. The goal was to provide a clear and accessible overview so that future researchers planning experiments using ASOs can easily engage with the topic.
The section “Design and Chemical Modifications of ASOs for Optimized Biological Effect” has been substantially improved. In particular, we have added a detailed description of the chemical modifications commonly used to enhance ASO stability, specificity, and efficacy. Additionally, the part addressing ASO design has been carefully revised and reorganised to emphasise the most important conclusions and practical considerations. These changes are intended to provide readers with a clearer, more structured overview of both the design principles and chemical strategies used to optimise ASO biological activity.
The section “Gene Expression Control by ASOs Utilising Naturally Occurring Mechanisms” has been significantly revised. In particular, we have added bullet points highlighting the main mechanisms of action. The section has also been substantially expanded and now, in our opinion, contains the most important information. The aim of this section is to familiarise readers with the various mechanisms through which ASOs can act, presented in a clear and accessible manner. Additionally, the section has been supplemented with entirely new and more precise illustrations to enhance clarity and understanding.
The section “Methods of ASO Delivery” has been revised primarily to include ASO delivery into animal cells. For plants, all currently known effective methods for ASO delivery have been included.
In the section “Applications of ASO Technology”, some references to experimental studies in plants are the same as those cited in our previous article, as we consider them highly important. However, additional references to other studies have also been included. Furthermore, the entire section has been significantly expanded to cover the applications of ASOs in animal cells as well as in medicine.
Comments/Recommendation 2: Line 433: the sentence «Application of Antisense Oligonucleotides as a Tool for Response: Specific Gene Silencing in Plant Research » seems to be a subtitle, but it is not highlighted as a subtitle.
Response 2: The aim was to create a single, unified section covering the applications of ASOs in both plants and animals. We did not intend to introduce an additional subsection in this part of the manuscript. This sentence was removed to improve the clarity of the text.
Comments/Recommendation 3: Lines 228-230: «Sequences with a GC content above 55% generally demonstrate stronger binding, while the presence of specific motifs—such as CTCT, GCCA, CCAC, TCCC, or ACTC—can further enhance duplex stability [4], [44] » authors should remove the reference for their previous review as it does not contain results of original study mentioned here. Alternatively, authors should rewrite and shorten the «Design» section with addition of reference «reviewed in [4]» because all the information was published before.
Response 3: The citation has been removed. We decided to retain the full description of ASO design, as the aim of this review is to compile all the most important information regarding ASO technology in a single article.
Comments/Recommendation 4: Line 251: «ASOs can also be directed to DNA sequences to induce epigenetic modifications, such as DNA methylation or chromatin remodeling [4]» please provide a link to original study but not the review.
Response 4: The citation has been updated.
Comments/Recommendation 5: Line 369: «Plant cells exhibit greater susceptibility to ASOs than animal cells, primarily due to their positively charged cell walls, which do not pose a barrier to the negatively charged ASO molecules [4]» please provide a link to original study.
Response 5: The section in the manuscript has been revised accordingly, and an updated citation has been included to reflect the changes.
Comments/Recommendation 6: Line 478: «Promising applications include the generation of epigenetically modified organisms (EMOs) as alternatives to GMOs, which remain tightly regulated in the EU [4].» please provide a link to studies demonstrating generation of epigenetically modified organisms, if possible, and a link to EU regulations.
Response 6: The citation has been updated
Comments/Recommendation 7: Lines 233-236: «Bioinformatic tools, such as mfold, enable the assessment of minimum free energy (ΔG) and potential RNA conformations, whereas sfold can identify transcript regions most favorable for hybridization.» here is a good sentence for reference for the previous review «reviewed in [4]» as it contains more detailed and useful information about the bioinformatic tools.
Response 7: The phrasing “More detailed information on this topic can be found in our previous review.” has been added to the text.
Round 2
Reviewer 3 Report
Comments and Suggestions for Authors
The authors took all the recommendations into account and improved the manuscript. I would be glad to see this article published.